

# The accuracy of Fiber-Optic Raman Spectroscopy in the detection and diagnosis of head and neck neoplasm *in vivo*: a systematic review and meta-analysis

Wen Chen[1], Yafei Chen[2], Chenzhou Wu[2], Xidong Zhang[3] and Xiaofeng Huang[1]

[1] Department of Stomatology and Immunology Research Center for Oral and Systemic Health, Beijing Friendship Hospital, Capital Medical University, Beijing, China
[2] State Key Laboratory of Oral Diseases & National Clinical Research Center for Oral Diseases, Department of Head and Neck Oncology, West China Hospital of Stomatology, Sichuan University, Chengdu, China
[3] Department of Pharmacy, The Fourth Hospital of Hebei Medical University, Shijiazhuang, China

Corresponding author
Xiaofeng Huang,
huangxf1998@163.com

## ABSTRACT

**Purpose**. The aim of this article was to review and collectively assess the published studies of fiber-optic Raman spectroscopy (RS) of the *in vivo* detection and diagnosis of head and neck carcinomas, and to derive a consensus average of the accuracy, sensitivity and specificity.

**Methods**. The authors searched four databases, including Ovid-Medline, Ovid-Embase, Cochrane Library, and the China National Knowledge Infrastructure (CNKI), up to February 2023 for all published studies that assessed the diagnostic accuracy of fiber-optic RS in the *in vivo* detection of head and neck carcinomas. Nonqualifying studies were screened out in accordance with the specified exclusion criteria, and relevant information about the diagnostic performance of fiber-optic RS was excluded. Publication bias was estimated by Deeks' funnel plot asymmetry test. A random effects model was adopted to calculate the pooled sensitivity, specificity and diagnostic odds ratio (DOR). Additionally, the authors conducted a summary receiver operating characteristic (SROC) curve analysis and threshold analysis, reporting the area under the curve (AUC) to evaluate the overall performance of fiber-optic RS *in vivo*.

**Results**. Ten studies (including 16 groups of data) were included in this article, and a total of 5,365 *in vivo* Raman spectra (cancer = 1,746; normal = 3,619) were acquired from 877 patients. The pooled sensitivity and specificity of fiber-optic RS of head and neck carcinomas were 0.88 and 0.94, respectively. SROC curves were generated to estimate the overall diagnostic accuracy, and the AUC was 0.96 (95% CI [0.94–0.97]). No significant publication bias was found in this meta-analysis by Deeks' funnel plot asymmetry test. The heterogeneity of these studies was significant; the $Q$ test values of the sensitivity and specificity were 106.23 ($P = 0.00$) and 64.21 ($P = 0.00$), respectively, and the I2 index of the sensitivity and specificity were 85.88 (95% CI [79.99–91.77]) and 76.64 (95% CI [65.45–87.83]), respectively.

**Conclusion**. Fiber-optic RS was demonstrated to be a reliable technique for the *in vivo* detection of head and neck carcinoma with high accuracy. However, considering

the high heterogeneity of these studies, more clinical studies are needed to reduce the heterogeneity, and further confirm the utility of fiber-optic Raman spectroscopy *in vivo*.

## INTRODUCTION

Malignant tumors are one of the main causes of death in humans. Worldwide, head and neck carcinomas are the sixth most common type of neoplasm, with approximately 940,000 new cases in 2018 (*Bray et al., 2018*; *Jemal et al., 2011*), and the major risk factors include tobacco, alcohol, human papilloma virus (HPV) and Epstein–Barr virus (EBV) (*Bray et al., 2018*; *Jemal et al., 2011*; *Ferlay et al., 2015*). Surfaced in the upper aerodigestive tract, including the oral cavity, pharynx, larynx, and paranasal sinuses, as well as cancers of the thyroid and major and minor salivary glands, were head and neck carcinomas (*Lydiatt et al., 2017*). In addition, squamous cell carcinoma makes up most of all head and neck cancers. Despite advances in the diagnosis and treatment of head and neck carcinomas, the 5-year survival rate is still under 50% worldwide, and this rate decreases to 19% for patients in the advanced stage of the disease (*Kumar, Abbas & Aster, 2010*). Early diagnosis and treatment of premalignant lesions and malignancies are crucial to minimize mortality and improve patient survival. However, current diagnostic techniques are often costly, invasive and time-consuming. Histological examination (HE) requires an invasive incision and usually takes 3–7 days (*Szybiak, Trzeciak & Golusiński, 2012*). Computerized tomography (CT) images and magnetic resonance imaging (MRI) are not sufficiently accurate and are prone to subjective explanation (*Zhan et al., 2020*). Thus, an accurate diagnostic technique with high efficiency for head and neck carcinomas is needed.

Raman spectroscopy (RS), an inelastic light scattering technique, is considered to be a promising diagnostic method. In the fingerprint (FP) range (*i.e.,* 800–1,800 cm$^{-1}$) and high-wavenumber (HW) (*i.e.,* 2,800–3,600 cm$^{-1}$) range, RS has the ability to reveal specific biochemical and biomolecular structures; therefore, it provides a unique opportunity to identify premalignant lesions and malignant tissue at the molecular level. Fiber-optic Raman spectroscopy has many applications, and it can be a modified technique for real-time *in vivo* detection, demonstrating superb diagnostic potential in clinical surroundings (*Lin et al., 2016a*; *Chen et al., 2018*; *Žuvela et al., 2019*).

To date, many studies have reported the accuracy of fiber-optic RS in the diagnosis of head and neck carcinomas, and some of these articles have focused on the accuracy of fiber-optic RS *in vivo*. However, no conclusion has been reached (*Žuvela et al., 2019*; *Lin et al., 2016b*; *Lin et al., 2017*; *Malik et al., 2017*; *Krishna et al., 2014*; *Lin, Cheng & Huang, 2012*; *Singh et al., 2013*; *Sahu et al., 2015*; *Ming et al., 2017*; *Lin et al., 2018*). In this meta-analysis, we aimed to systematically assess the diagnostic accuracy of fiber-optic RS in the rapid discrimination of head and neck carcinomas.

## MATERIAL AND METHODS

### Search strategy

All studies were identified by systematically searching OVID EMBASE, OVID MEDLINE, Cochrane Library, and CNKI databases (up to February 2023), and there was no limit to the start date of the search. In this study Wen Chen, Yafei Chen and Chenzhou Wu performed the search strategy.

The authors display the details of the search strategy in Table 1.

### Selection criteria

Studies were evaluated on the basis of the following criteria for inclusion: (1) only *in vivo* human samples of head and neck carcinomas were detected and diagnosed by fiber-optic RS. (2) All samples with head and neck carcinomas were investigated with histopathological diagnosis as the gold standard. (3) A healthy control group without head and neck carcinomas was included in the studies. (4) Data in the article can be used to construct a fourfold table including true positives (TPs), true negatives (TNs), false positives (FPs) and false negatives (FNs).

The exclusion criteria were as follows: (1) *ex vivo* sample detected, (2) studies that did not have a control group, and (3) reviews or duplicate reports.

### Data extraction

We downloaded the full texts of all potential studies to ensure that they were eligible for inclusion. Three reviewers (Wen Chen, Yafei Chen and Chenzhou Wu) independently screened the 324 articles (title/abstract and full text). The whole screening process is blinded and the text software is used. Two reviewers independently extracted the data of each article and evaluated the quality of the article utilizing a standardized data extraction form. Disagreements were resolved by consensus. Data were collected as previously described in *Zhan et al. (2020)*, specifically the first author's name, geographical location, demographic data (participants' age and sex), tumor position, sample type, diagnostic algorithm, spectroscopy range, acquisition time, TP, TN, FP and FN.

### Statistical analysis

All meta-analyses were performed in Stata 15.1 (Stata Corp, College Station, TX, USA).

The sensitivity, specificity, diagnostic threshold, diagnostic odds ratio (DOR), and 95% confidence interval (CI) were calculated to obtain the diagnostic accuracy of fiber-optic RSfor head and neck carcinomas. Outcome data were subject to statistical pooling through random effect models, which suggests that the studies from populations may affect the final results (*Melsen et al., 2014*; *Lean et al., 2009*). Also, we used the midas module to calculate summary statistics and SROC. The commands were "midas TP, TN, FP, FN, res(all)" and "midas TP, TN, FP, FN, plot sroc(both)", respectively.

A summary receiver operating characteristic (SROC) curve and threshold analysis were carried out to investigate the threshold. The area under the curve (AUC) was calculated to evaluate the overall effectiveness of fiber-optic RS. If the SROC curves exhibited a shoulder peak, it indicated that thresholds may have an impact on the result. The diagnostic effect

**Table 1  Search strategies in the study.** Search strategies used in this article.

| Databases | Steps | Strategies |
|---|---|---|
| | #1 | (head and neck neoplasms).mp. [mp=title, abstract, original title, name of substance word, subject heading word, floating sub-heading word, keyword heading word, organism supplementary concept word, protocol supplementary concept word, rare disease supplementary concept word, unique identifier, synonyms] |
| | #2 | Facial Neoplasms.mp. or Facial Neoplasms/ |
| | #3 | Eyelid Neoplasms.mp. or Eyelid Neoplasms/ |
| | #4 | Mouth Neoplasms.mp. or Mouth Neoplasms/ |
| | #5 | Gingival Neoplasms.mp. or Gingival Neoplasms/ |
| | #6 | Leukoplakia, Oral.mp. or Leukoplakia, Oral/ |
| | #7 | Leukoplakia, Hairy.mp. or Leukoplakia, Hairy/ |
| | #8 | Lip Neoplasms.mp. or Lip Neoplasms/ |
| | #9 | Palatal Neoplasms.mp. or Palatal Neoplasms/ |
| | #10 | Salivary Gland Neoplasms.mp. or Salivary Gland Neoplasms/ |
| | #11 | Parotid Neoplasms.mp. or Parotid Neoplasms/ |
| | #12 | Sublingual Gland Neoplasms.mp. or Sublingual Gland Neoplasms/ |
| | #13 | Submandibular Gland Neoplasms.mp. or Submandibular Gland Neoplasms/ |
| | #14 | Tongue Neoplasms.mp. or Tongue Neoplasms/ |
| | #15 | Otorhinolaryngologic Neoplasms.mp. or Otorhinolaryngologic Neoplasms/ |
| | #16 | Otorhinolaryngologic Neoplasms.mp. or Otorhinolaryngologic Neoplasms/ |
| | #17 | Laryngeal Neoplasms.mp. or Laryngeal Neoplasms/ |
| | #18 | Nose Neoplasms.mp. or Nose Neoplasms/ |
| | #19 | Paranasal Sinus Neoplasms.mp. or Paranasal Sinus Neoplasms/ |
| | #20 | Maxillary Sinus Neoplasms.mp. or Maxillary Sinus Neoplasms/ |
| | #21 | Pharyngeal Neoplasms.mp. or Pharyngeal Neoplasms/ |
| | #22 | Hypopharyngeal Neoplasms.mp. or Hypopharyngeal Neoplasms/ |
| | #23 | Nasopharyngeal Neoplasms.mp. or Nasopharyngeal Neoplasms/ |
| | #24 | Nasopharyngeal Carcinoma.mp. or Nasopharyngeal Neoplasms/ or Nasopharyngeal Carcinoma/ |
| | #25 | Oropharyngeal Neoplasms.mp. or Oropharyngeal Neoplasms/ |
| | #26 | Tonsillar Neoplasms.mp. or Tonsillar Neoplasms/ |
| | #27 | Parathyroid Neoplasms.mp. or Parathyroid Neoplasms/ |
| | #28 | (Squamous Cell Carcinoma of Head and Neck).mp. [mp=title, abstract, original title, name of substance word, subject heading word, floating sub-heading word, keyword heading word, organism supplementary concept word, protocol supplementary concept word, rare disease supplementary concept word, unique identifier, synonyms] |
| | #29 | Thyroid Neoplasms.mp. or Thyroid Neoplasms/ |
| | #30 | Thyroid Cancer, Papillary.mp. or Thyroid Cancer, Papillary/ |
| | #31 | Thyroid Nodule.mp. or Thyroid Nodule/ |
| | #32 | Tracheal Neoplasms.mp. or Tracheal Neoplasms/ |
| | #33 | (Neoplasms, Head and Neck).mp. [mp=title, abstract, original title, name of substance word, subject heading word, floating sub-heading word, keyword heading word, organism supplementary concept word, protocol supplementary concept word, rare disease supplementary concept word, unique identifier, synonyms] |
| | #34 | Head, Neck Neoplasms.mp. or ''Head and Neck Neoplasms''/ |

**Table 1** (*continued*)

| Databases | Steps | Strategies |
|---|---|---|
| | #35 | (Cancer of Head and Neck).mp. [mp=title, abstract, original title, name of substance word, subject heading word, floating sub-heading word, keyword heading word, organism supplementary concept word, protocol supplementary concept word, rare disease supplementary concept word, unique identifier, synonyms] |
| | #36 | (Head and Neck Cancer).mp. [mp=title, abstract, original title, name of substance word, subject heading word, floating sub-heading word, keyword heading word, organism supplementary concept word, protocol supplementary concept word, rare disease supplementary concept word, unique identifier, synonyms] |
| | #37 | (Cancer of the Head and Neck).mp. [mp=title, abstract, original title, name of substance word, subject heading word, floating sub-heading word, keyword heading word, organism supplementary concept word, protocol supplementary concept word, rare disease supplementary concept word, unique identifier, synonyms] |
| OVID-Medline (204 studies) | #38 | Head Neoplasms.mp. or "Head and Neck Neoplasms"/ |
| | #39 | Neoplasms, Head.mp. or "Head and Neck Neoplasms"/ |
| | #40 | Neck Neoplasms.mp. or "Head and Neck Neoplasms"/ |
| | #41 | Neoplasms, Neck.mp. or "Head and Neck Neoplasms"/ |
| | #42 | Cancer of Head.mp. or "Head and Neck Neoplasms"/ |
| | #43 | Head Cancer.mp. or "Head and Neck Neoplasms"/ |
| | #44 | Cancer of the Head.mp. or "Head and Neck Neoplasms"/ |
| | #45 | Cancer of Neck.mp. or "Head and Neck Neoplasms"/ |
| | #46 | Neck Cancer.mp. or "Head and Neck Neoplasms"/ |
| | #47 | Cancer of the Neck.mp. or "Head and Neck Neoplasms"/ |
| | #48 | ((nasopharyn$ or oropharyn$ or laryn$ or glotti$ or tonsil$ or epiglotti$ or oral cavity or oral or tongue or gingiva$ or bucca$ or lip or palat$ or gum or mouth floor or floor of mouth or lingual or (head and neck) or HN) adj4 (cancer$ or tumor$ or tumour$ or neoplasm$ or carcinoma$ or squamous cell carcinoma or SCC)).mp. |
| | #49 | ((cancer$ or tumor$ or tumour$ or neoplasm$ or carcinoma$ or squamous cell carcinoma or SCC) adj4 (nasopharyn$ or oropharyn$ or laryn$ or glotti$ or tonsil$ or epiglotti$ or oral cavity or oral or tongue or gingiva$ or bucca$ or lip or palat$ or gum or mouth floor or floor of mouth or lingual or (head and neck) or HN)).mp. |
| | #50 | (HNSCC or SCCHN or HNC or OSCC or OCSCC or OPSCC or LSCC or NPC).mp. |
| | #51 | spectrum analysis,raman.mp. or Spectrum Analysis, Raman/ |
| | #52 | Raman Spectrum Analysis.mp. or Spectrum Analysis, Raman/ |
| | #53 | Raman Spectroscopy.mp. or Spectrum Analysis, Raman/ |
| | #54 | Spectroscopy, Raman.mp. or Spectrum Analysis, Raman/ |
| | #55 | Analysis, Raman Spectrum.mp. or Spectrum Analysis, Raman/ |
| | #56 | Raman Optical Activity Spectroscopy.mp. or Spectrum Analysis, Raman/ |
| | #57 | Raman Scattering.mp. or Spectrum Analysis, Raman/ |
| | #58 | Scattering, Raman.mp. or Spectrum Analysis, Raman/ |
| | #59 | #1 or #2 or #3 or #4 or #5 or #6 or #7 or #8 or #9 or #10 or #11 or #12 or #13 or #14 or #15 or #16 or #17 or #18 or #19 or #20 or #21 or #22 or #23 or #24 or #25 or #26 or #27 or #28 or #29 or #30 or #31 or #32 or #33 or #34 or #35 or #36 or #37 or #38 or #39 or #40 or #41 or #42 or #43 or #44 or #45 or #46 or #47 or #48 or #49 or #50 or |
| | #60 | #51 or #52 or #53 or #54 or #55 or #56 or #57 or #58 |
| | #61 | #59 and #60 |

**Table 1** (*continued*)

| Databases | Steps | Strategies |
|---|---|---|
| | #1 | (head and neck neoplasms).mp. [mp=title, abstract, heading word, drug trade name, original title, device manufacturer, drug manufacturer, device trade name, keyword, floating subheading word, candidate term word] |
| | #2 | Facial Neoplasms.mp. or Facial Neoplasms/ |
| | #3 | Eyelid Neoplasms.mp. or Eyelid Neoplasms/ |
| | #4 | Mouth Neoplasms.mp. or Mouth Neoplasms/ |
| | #5 | Gingival Neoplasms.mp. or Gingival Neoplasms/ |
| | #6 | Leukoplakia, Oral.mp. or Leukoplakia, Oral/ |
| | #7 | Leukoplakia, Hairy.mp. or Leukoplakia, Hairy/ |
| | #8 | Lip Neoplasms.mp. or Lip Neoplasms/ |
| | #9 | Palatal Neoplasms.mp. or Palatal Neoplasms/ |
| | #10 | Salivary Gland Neoplasms.mp. or Salivary Gland Neoplasms/ |
| | #11 | Parotid Neoplasms.mp. or Parotid Neoplasms/ |
| | #12 | Sublingual Gland Neoplasms.mp. or Sublingual Gland Neoplasms/ |
| | #13 | Submandibular Gland Neoplasms.mp. or Submandibular Gland Neoplasms/ |
| | #14 | Tongue Neoplasms.mp. or Tongue Neoplasms/ |
| | #15 | Otorhinolaryngologic Neoplasms.mp. or Otorhinolaryngologic Neoplasms/ |
| | #16 | Ear Neoplasms.mp. or Ear Neoplasms/ |
| | #17 | Laryngeal Neoplasms.mp. or Laryngeal Neoplasms/ |
| | #18 | Nose Neoplasms.mp. or Nose Neoplasms/ |
| | #19 | Paranasal Sinus Neoplasms.mp. or Paranasal Sinus Neoplasms/ |
| | #20 | Maxillary Sinus Neoplasms.mp. or Maxillary Sinus Neoplasms/ |
| | #21 | Pharyngeal Neoplasms.mp. or Pharyngeal Neoplasms/ |
| | #22 | Hypopharyngeal Neoplasms.mp. or Hypopharyngeal Neoplasms/ |
| | #23 | Nasopharyngeal Neoplasms.mp. or Nasopharyngeal Neoplasms/ |
| | #24 | Nasopharyngeal Carcinoma.mp. or Nasopharyngeal Neoplasms/ or Nasopharyngeal Carcinoma/ |
| | #25 | Oropharyngeal Neoplasms.mp. or Oropharyngeal Neoplasms/ |
| | #26 | Tonsillar Neoplasms.mp. or Tonsillar Neoplasms/ |
| | #27 | Parathyroid Neoplasms.mp. or Parathyroid Neoplasms/ |
| | #28 | (Squamous Cell Carcinoma of Head and Neck).mp. [mp=title, abstract, heading word, drug trade name, original title, device manufacturer, drug manufacturer, device trade name, keyword, floating subheading word, candidate term word] |
| | #29 | Thyroid Neoplasms.mp. or Thyroid Neoplasms/ |
| | #30 | Thyroid Cancer, Papillary.mp. or Thyroid Cancer, Papillary/ |
| | #31 | Thyroid Nodule.mp. or Thyroid Nodule/ |
| | #32 | Tracheal Neoplasms.mp. or Tracheal Neoplasms/ |
| | #33 | (Neoplasms, Head and Neck).mp. [mp=title, abstract, heading word, drug trade name, original title, device manufacturer, drug manufacturer, device trade name, keyword, floating subheading word, candidate term word] |
| | #34 | Head, Neck Neoplasms.mp. or ''Head and Neck Neoplasms''/ |
| | #35 | (Cancer of Head and Neck).mp. [mp=title, abstract, heading word, drug trade name, original title, device manufacturer, drug manufacturer, device trade name, keyword, floating subheading word, candidate term word] |

**Table 1** (*continued*)

| Databases | Steps | Strategies |
|---|---|---|
| Embase (285 studies) | #36 | (Head and Neck Cancer).mp. [mp=title, abstract, heading word, drug trade name, original title, device manufacturer, drug manufacturer, device trade name, keyword, floating subheading word, candidate term word] |
| | #37 | (Cancer of the Head and Neck).mp. [mp=title, abstract, heading word, drug trade name, original title, device manufacturer, drug manufacturer, device trade name, keyword, floating subheading word, candidate term word] |
| | #38 | Head Neoplasms.mp. or "Head and Neck Neoplasms"/ |
| | #39 | Neoplasms, Head.mp. or "Head and Neck Neoplasms"/ |
| | #40 | Neck Neoplasms.mp. or "Head and Neck Neoplasms"/ |
| | #41 | Neoplasms, Neck.mp. or "Head and Neck Neoplasms"/ |
| | #42 | Cancer of Head.mp. or "Head and Neck Neoplasms"/ |
| | #43 | Head Cancer.mp. or "Head and Neck Neoplasms"/ |
| | #44 | Cancer of the Head.mp. or "Head and Neck Neoplasms"/ |
| | #45 | Cancer of Neck.mp. or "Head and Neck Neoplasms"/ |
| | #46 | Neck Cancer.mp. or "Head and Neck Neoplasms"/ |
| | #47 | Cancer of the Neck.mp. or "Head and Neck Neoplasms"/ |
| | #48 | ((nasopharyn$ or oropharyn$ or laryn$ or glotti$ or tonsil$ or epiglotti$ or oral cavity or oral or tongue or gingiva$ or bucca$ or lip or palat$ or gum or mouth floor or floor of mouth or lingual or (head and neck) or HN) adj4 (cancer$ or tumor$ or tumour$ or neoplasm$ or carcinoma$ or squamous cell carcinoma or SCC)).mp. |
| | #49 | ((cancer$ or tumor$ or tumour$ or neoplasm$ or carcinoma$ or squamous cell carcinoma or SCC) adj4 (nasopharyn$ or oropharyn$ or laryn$ or glotti$ or tonsil$ or epiglotti$ or oral cavity or oral or tongue or gingiva$ or bucca$ or lip or palat$ or gum or mouth floor or floor of mouth or lingual or (head and neck) or HN)).mp. |
| | #50 | (HNSCC or SCCHN or HNC or OSCC or OCSCC or OPSCC or LSCC or NPC).mp. |
| | #51 | spectrum analysis,raman.mp. or Spectrum Analysis, Raman/ |
| | #52 | Raman Spectrum Analysis.mp. or Spectrum Analysis, Raman/ |
| | #53 | Raman Spectroscopy.mp. or Spectrum Analysis, Raman/ |
| | #54 | Spectroscopy, Raman.mp. or Spectrum Analysis, Raman/ |
| | #55 | Analysis, Raman Spectrum.mp. or Spectrum Analysis, Raman/ |
| | #56 | Raman Optical Activity Spectroscopy.mp. or Spectrum Analysis, Raman/ |
| | #57 | Raman Scattering.mp. or Spectrum Analysis, Raman/ |
| | #58 | Scattering, Raman.mp. or Spectrum Analysis, Raman/ |
| | #59 | #1 or #2 or #3 or #4 or #5 or #6 or #7 or #8 or #9 or #10 or #11 or #12 or #13 or #14 or #15 or #16 or #17 or #18 or #19 or #20 or #21 or #22 or #23 or #24 or #25 or #26 or #27 or #28 or #29 or #30 or #31 or #32 or #33 or #34 or #35 or #36 or #37 or #38 or #39 or #40 or #41 or #42 or #43 or #44 or #45 or #46 or #47 or #48 or #49 or #50 |
| | #60 | #51 or #52 or #53 or #54 or #55 or #56 or #57 or #58 |
| | #61 | #59 and #60 |
| Cochrane library(2 studies) | #1 | MeSH descriptor: [Spectrum Analysis, Raman] explode all trees |
| | #2 | MeSH descriptor: [Neoplasms] explode all trees |
| | #3 | #1 and #2 |

**Table 1** (*continued*)

| Databases | Steps | Strategies |
|---|---|---|
| CNKI(16 studies) | #1 | head and neck neoplasms |
| | #2 | mouth neoplasms |
| | #3 | Nasopharyngeal Neoplasms |
| | #4 | Laryngeal Neoplasms |
| | #5 | Raman |
| | #6 | #1 or #2 or #3 or #4 |
| | #7 | #5 and #6 |

**Notes.**

The data in this table is up to February 2023.

was excellent when the AUC value was between 0.9 and 1, favorable when the AUC value was between 0.8 and 0.9, fair when the AUC value was between 0.7 and 0.8, and poor when the AUC value was between 0.6 and 0.7. The diagnostic method was considered to have failed when the AUC fell between 0.5 and 0.6 (*Metz, 1978*).

The $Q$ statistic and the inconsistency index ($I^2$) statistic were used to further investigate heterogeneity. The Q statistic was used to illustrate the presence or absence of heterogeneity, and the $I^2$ index was used to classify the degree of heterogeneity (*Huedo-Medina et al., 2006*). The degree of heterogeneity was considered to be significant when the $I^2$ index was greater than 50% and the $P$ value was less than 0.05 (*Higgins et al., 2003*). Subgroup analyses were performed for substantial heterogeneity. Publication bias was estimated by Deeks' funnel plot asymmetry test, which was considered to exist when the $P$ value was less than 0.05 (*Begg & Mazumdar, 1994*).

## Quality assessment

The Quality Assessment of Diagnostic Accuracy Studies (QUADAS-2) guidelines were used to systematically assess the quality of the studies included in this meta-analysis (high, unclear, or low) (*Whiting et al., 2011*). The main items included (1) patient selection, (2) the index test, (3) the reference standard and (4) flow and timing. The risk of bias was rated as low risk, high risk or unclear risk. The QUADAS-2 was performed by Review Manager 5.4. The quality of the included studies was evaluated independently by two reviewers (Yafei Chen and Chenzhou Wu) according to the QUADAS-2 guidelines. Disagreements were resolved by a third reviewer (Wen Chen).

# RESULTS

## Study selection and description of studies included in the article

Initially, the authors searched 658 articles from OVID EMBASE, OVID MEDLINE, the Cochrane Library and CNKI databases. After removing duplicates, 324 articles were selected. Then, 86 articles were identified after screening the titles and abstracts. Finally, 10 eligible articles were included in this meta-analysis. The full study screening and selection process is presented in Fig. 1.

Sixteen groups of data from 10 articles were extracted from these articles due to their characteristics (such as different spectrum ranges and different acquisition times), and

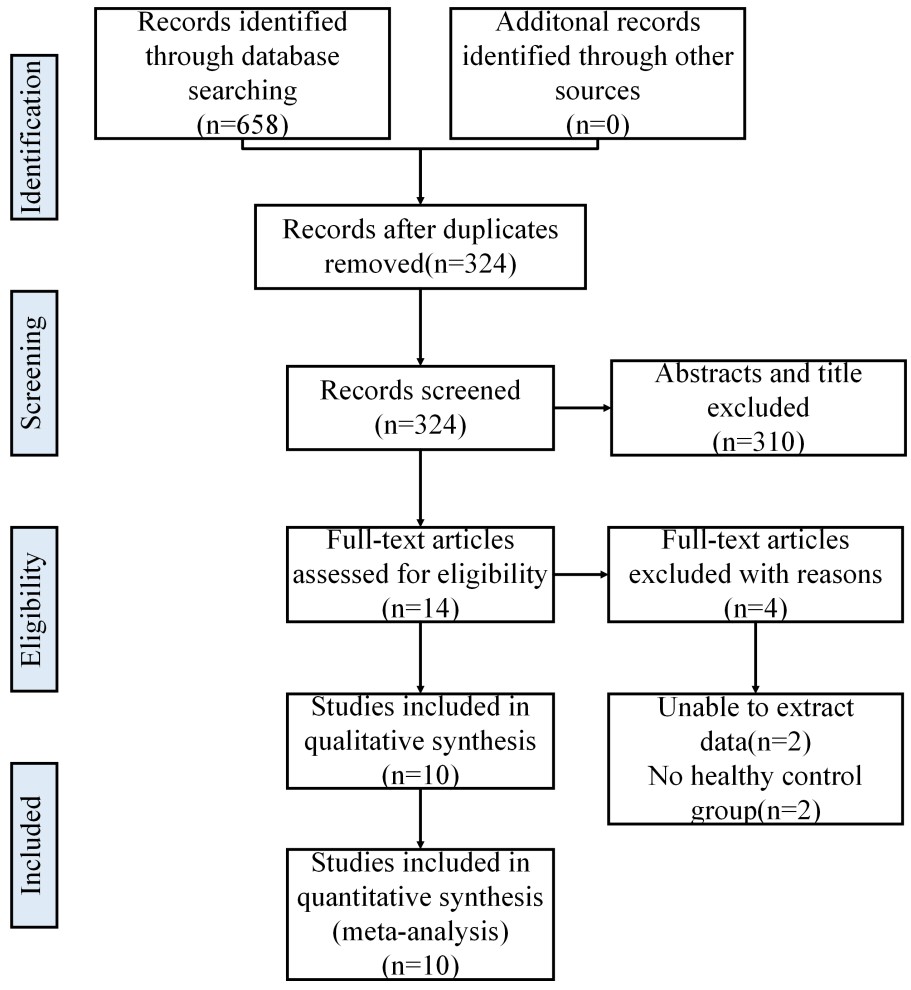

**Figure 1  Literature search and selection.**

none of these 16 groups of data were duplicated. Of all eligible studies (*Žuvela et al., 2019*; *Lin et al., 2016b*; *Lin et al., 2017*; *Malik et al., 2017*; *Krishna et al., 2014*; *Lin, Cheng & Huang, 2012*; *Singh et al., 2013*; *Sahu et al., 2015*; *Ming et al., 2017*; *Lin et al., 2018*), all articles were written in English. Among the 16 groups of data, the sample types included the larynx ($n = 4$) (*Lin et al., 2016b*; *Lin, Cheng & Huang, 2012*), the nasopharynx ($n = 8$) (*Žuvela et al., 2019*; *Lin et al., 2017*; *Ming et al., 2017*; *Lin et al., 2018*) and the oral cavity ($n = 4$) (*Malik et al., 2017*; *Krishna et al., 2014*; *Singh et al., 2013*; *Sahu et al., 2015*). The Raman spectral range applied in all eligible studies was divided into three categories, including the fingerprint region (FP) ($n = 8$) (*Žuvela et al., 2019*; *Lin et al., 2016b*; *Lin et al., 2017*; *Malik et al., 2017*; *Krishna et al., 2014*; *Singh et al., 2013*; *Sahu et al., 2015*; *Lin et al., 2018*), high wavenumber (HW) ( $n = 4$) (*Žuvela et al., 2019*; *Lin et al., 2016b*; *Lin et al., 2017*; *Lin, Cheng & Huang, 2012*) and FP + HW ($n = 4$) (*Žuvela et al., 2019*; *Lin et al., 2016b*; *Lin et al., 2017*; *Ming et al., 2017*). Considering the variable Raman instrumentation used in the ten studies, acquisition times were divided into two groups: acquisition times
$\leq 1$ s ($n = 12$) (*Žuvela et al., 2019*; *Lin et al., 2016b*; *Lin et al., 2017*; *Lin, Cheng & Huang, 2012*; *Ming et al., 2017*; *Lin et al., 2018*) and acquisition times $> 1$ s ($n = 4$) (*Malik et al., 2017*; *Krishna et al., 2014*; *Singh et al., 2013*; *Sahu et al., 2015*). The details of each group of data are shown in Table 2. Partial least squares-discrimination analysis (PLS-DA), Leave-one-out cross-validation (LOOCV), principal component analysis + Linear discriminant analysis (PCA + LDA), genetic algorithm-partial least squares-linear discriminant analysis (GA-PLS-LDA), and stepwise analysis of multiple linear regression (SMLR) in Table 2 refer to different diagnostic algorithms of Raman spectra.

## Pooled results
### Overall results

Ten studies (*Žuvela et al., 2019*; *Lin et al., 2016b*; *Lin et al., 2017*; *Malik et al., 2017*; *Krishna et al., 2014*; *Lin, Cheng & Huang, 2012*; *Singh et al., 2013*; *Sahu et al., 2015*; *Ming et al., 2017*; *Lin et al., 2018*) (16 groups of data) were included. In these studies, a total of 5365 *in vivo* Raman spectra (cancer = 1,746; normal = 3,619) were acquired from 877 patients. Their coalescent sensitivity and specificity results for fiber-optic RS were 0.88 (95% CI [0.84–0.91], $P = 0.00$, $I^2 = 85.88$) and 0.94 (95% CI [0.91–0.96], $P = 0.00$, $I^2 = 76.64$), respectively. The DOR was 105.69 (95% CI [67.50–165.47], $P = 0.00$, $I^2 = 100.00$). SROC curves were generated to estimate the overall diagnostic accuracy, and the AUC was 0.96 (95% CI [0.94–0.97]).

## Site of disease
### Larynx cancer

Two of the included studies (*Lin et al., 2016b*; *Lin, Cheng & Huang, 2012*) (4 groups of data) assessed larynx samples. A total of 397 *in vivo* Raman spectra (cancer = 162; normal = 235) were acquired from 99 patients. The coalescent sensitivity and specificity results for fiber-optic RS were 0.88 (95% CI [0.81–0.92], $P = 0.19$, $I^2 = 36.52$) and 0.88 (95% CI [0.83–0.92], $P = 0.85$, $I^2 = 0.00$), respectively. The DOR was 47.63 (95% CI [21.93–103.45], $P = 0.275$, $I^2 = 22.6$). SROC curves were generated to estimate the overall diagnostic accuracy, and the AUC was 0.92 (95% CI [0.89–0.94]).

### Nasopharyngeal cancer

Four of the included studies (*Žuvela et al., 2019*; *Lin et al., 2017*; *Ming et al., 2017*; *Lin et al., 2018*) (8 groups of data) assessed nasopharynx samples. A total of 1298 *in vivo* Raman spectra (cancer = 580; normal = 718) were acquired from 296 patients. Their coalescent sensitivity and specificity results for fiber-optic RS were 0.88 (95% CI [0.83–0.91], $P = 0.02$, $I^2 = 57.83$) and 0.97 (95% CI [0.90–0.99], $P = 0.00$, $I^2 = 72.94$), respectively. The DOR was 118.07 (95% CI [71.38–195.30], $P = 0.294$, $I^2 = 17.3$). SROC curves were generated to estimate the overall diagnostic accuracy. The AUC was 0.94 (95% CI [0.92–0.96]).

### Oral cancer

Four of the included studies (*Malik et al., 2017*; *Krishna et al., 2014*; *Singh et al., 2013*; *Sahu et al., 2015*) (4 groups of data) assessed oral samples. A total of 3670 *in vivo* Raman spectra (cancer = 1,004; normal = 2,666) were acquired from 482 patients. Their coalescent

Chen et al. (2023), *PeerJ*, DOI 10.7717/peerj.16536

**Table 2  General information of the studies included in the article.**

| Year | Author | Country | Disease | Number of people | Mean age | Male: female | Position | Sample type | Diagnostic algorithm | Spectrum range | TP | FN | TN | FP | Acquisition time |
|---|---|---|---|---|---|---|---|---|---|---|---|---|---|---|---|
| 2016 | Lin, K. a | Singapore | Laryngeal carcinoma | 60 | 51 | 47:13 | larynx | *in vivo* | PLS-DA + LOOCV | FP + HW | 28 | 2 | 64 | 7 | <0.2s |
| 2016 | Lin, K. b | Singapore | Laryngeal carcinoma | 60 | 51 | 47:13 | larynx | *in vivo* | PLS-DA + LOOCV | FP | 26 | 4 | 61 | 10 | <0.2s |
| 2016 | Lin, K. c | Singapore | Laryngeal carcinoma | 60 | 51 | 47:13 | larynx | *in vivo* | PLS-DA + LOOCV | HW | 23 | 7 | 62 | 9 | <0.2s |
| 2012 | Lin, K. | Singapore | Laryngeal carcinoma | 39 | 60 | –* | larynx | *in vivo* | PCA + LDA + LOOCV | HW | 65 | 7 | 20 | 2 | <1s |
| 2018 | Lin, D. | China | nasopharyngeal carcinoma | 60 | 53.8 | 39:21 | nasopharynx | *in vivo* | PCA + LDA | FP | 126 | 15 | 131 | 11 | 1s |
| 2019 | Zuvela, P. a | Singapore | nasopharyngeal carcinoma | 62 | Male 53.8 ± 16.9, Female 46.4 ± 11.3 | 43:19 | nasopharynx | *in vivo* | GA-PLS-LDA,LOOCV | FP + HW | 28 | 2 | 83 | 0 | <0.5s |
| 2019 | Zuvela, P. b | Singapore | nasopharyngeal carcinoma | 62 | Male 53.8 ± 16.9, Female 46.4 ± 11.3 | 43:19 | nasopharynx | *in vivo* | GA-PLS-LDA,LOOCV | FP | 21 | 9 | 83 | 0 | <0.5s |
| 2019 | Zuvela, P. c | Singapore | nasopharyngeal carcinoma | 62 | Male 53.8 ± 16.9, Female 46.4 ± 11.3 | 43:19 | nasopharynx | *in vivo* | GA-PLS-LDA,LOOCV | HW | 24 | 6 | 83 | 0 | <0.5s |
| 2017 | Lin, K. a | Singapore | nasopharyngeal carcinoma | 95 | 52 | 68:27 | nasopharynx | *in vivo* | PCA,LDA,LOOCV | FP + HW | 102 | 7 | 88 | 7 | <0.5s |
| 2017 | Lin, K. b | Singapore | nasopharyngeal carcinoma | 95 | 52 | 68:27 | nasopharynx | *in vivo* | PCA,LDA,LOOCV | FP | 98 | 11 | 84 | 11 | <0.5s |
| 2017 | Lin, K. c | Singapore | nasopharyngeal carcinoma | 95 | 52 | 68:27 | nasopharynx | *in vivo* | PCA,LDA,LOOCV | HW | 97 | 12 | 86 | 9 | <0.5s |
| 2017 | Ming, L.C. | Singapore | nasopharyngeal carcinoma | 79 | –[a] | 56:23 | nasopharynx | *in vivo* | PLS | FP + HW | 20 | 2 | 40 | 2 | 0.1s–0.5s |
| 2013 | Singh, S.P. | India | oral carcinoma | 84 | 46.3 | 75:9 | buccal | *in vivo* | PC-LDA,PCA,LDA,LOOCV | FP | 166 | 26 | 449 | 21 | 3s |
| 2014 | Krishna, H. | India | oral carcinoma | 199 | 41.2 | 6:1 | oral | *in vivo* | SMLR,LOOCV | FP | 281 | 35 | 458 | 28 | 5s |
| 2016 | Sahu, A. | India | oral carcinoma | 157 | 43 | 125:32 | oral | *in vivo* | PC-LDA,LOOCV,LDA | FP | 174 | 77 | 1246 | 106 | 3s |
| 2017 | Malik, A. | India | oral carcinoma | 42 | –* | –* | buccal | *in vivo* | PC-LDA,LOOCV | FP | 233 | 12 | 317 | 41 | 3s |

**Notes.**

[a] "–" in this table means no relative data in article was found.

Partial Least Squares-Discrimination Analysis (PLS-DA), Leave-one-out cross-validation (LOOCV), Principal component analysis + Linear discriminant analysis (PCA + LDA), Genetic algorithm—Partial Least Squares—Linear discriminant analysis (GA-PLS-LDA) and Stepwise analysis of multiple linear regression (SMLR) in Table 2 refer to different diagnostic algorithms of Raman spectra. Data in articles can be used to construct a fourfold table including true positives (TPs), true negatives (TNs), false positives (FPs) and false negatives (FNs).
sensitivity and specificity results for fiber-optic RS were 0.87 (95% CI [0.76–0.94], $P = 0.00$, $I^2 = 95.94$) and 0.93 (95% CI [0.90–0.95], $P = 0.00$, $I^2 = 88.50$), respectively. The DOR was 90.13 (95% CI [32.91–246.86], $P = 0.000$, $I^2 = 93.6$). SROC curves were generated to estimate the overall diagnostic accuracy, and the AUC was 0.96 (95% CI [0.94–0.97]).

## Raman spectral range
### Fingerprint range
Eight of the included studies (8 groups of data) (*Žuvela et al., 2019*; *Lin et al., 2016b*; *Lin et al., 2017*; *Malik et al., 2017*; *Krishna et al., 2014*; *Singh et al., 2013*; *Sahu et al., 2015*; *Lin et al., 2018*) assessed FP samples. A total of 4371 *in vivo* Raman spectra (cancer = 1,314; normal = 3,057) were acquired from 325 patients. Their coalescent sensitivity and specificity results for fiber-optic RS were 0.87 (95% CI [0.80–0.91], $P = 0.00$, $I^2 = 91.70$) and 0.93 (95% CI [0.90–0.95], $P = 0.00$, $I^2 = 81.45$), respectively. The DOR was 85.36 (95% CI [43.75–166.55], $P = 0.000$, $I^2 = 86.1$). SROC curves were generated to estimate the overall diagnostic accuracy. The AUC was 0.96 (95% CI [0.94–0.97]).

### High wavenumber range
Four of the included studies (4 groups of data) (*Žuvela et al., 2019*; *Lin et al., 2016b*; *Lin et al., 2017*; *Lin, Cheng & Huang, 2012*) assessed HW samples. A total of 512 *in vivo* Raman spectra (cancer = 241; normal = 271) were acquired from 256 patients. The coalescent sensitivity and specificity results for fiber-optic RSwere 0.86 (95% CI [0.81–0.91], $P = 0.17$, $I^2 = 41.00$) and 0.94 (95% CI [0.82–0.98], $P = 0.01$, $I^2 = 71.81$), respectively. The DOR was 66.59 (95% CI [24.19–183.29], $P = 0.102$, $I^2 = 51.7$). SROC curves were generated to estimate the overall diagnostic accuracy, and the AUC was 0.91 (95% CI [0.88–0.93]).

### Fingerprint range + high wavenumber range
Four of the included studies (4 groups of data) (*Lin et al., 2016b*; *Lin et al., 2017*; *Ming et al., 2017*; *Lin et al., 2018*) assessed FP + HW samples. A total of 482 *in vivo* Raman spectra (cancer = 191; normal = 291) were acquired from 296 patients. The coalescent sensitivity and specificity results for fiber-optic RSwere 0.93 (95% CI [0.88–0.96], $P = 0.97$, $I^2 = 0.00$) and 0.96 (95% CI [0.88–0.98], $P = 0.04$, $I^2 = 63.60$), respectively. The DOR was 199.73 (95% CI [89.96–443.45], $P = 0.493$, $I^2 = 0.0$). SROC curves were generated to estimate the overall diagnostic accuracy, and the AUC was 0.94 (95% CI [0.91–0.96]).

## Acquisition time
≤1 s
Six of the included studies (*Žuvela et al., 2019*; *Lin et al., 2016b*; *Lin et al., 2017*; *Lin, Cheng & Huang, 2012*; *Ming et al., 2017*; *Lin et al., 2018*) (12 groups of data) assessed acquisition times ≤1s. A total of 1,695 *in vivo* Raman spectra (cancer = 742; normal = 953) were acquired from 395 patients. Their coalescent sensitivity and specificity results for fiber-optic RSwere 0.88 (95% CI [0.85–0.91], $P = 0.03$, $I^2 = 49.80$) and 0.95 (95% CI [0.90–0.97], $P = 0.00$, $I^2 = 67.04$), respectively. The DOR was 92.21 (95% CI [56.49–150.52], $P = 0.101$, $I^2 = 36.2$). SROC curves were generated to estimate the overall diagnostic accuracy, and the AUC was 0.94 (95% CI [0.91–0.95]).

>1 s

Four of the included studies (*Malik et al., 2017*; *Krishna et al., 2014*; *Singh et al., 2013*; *Sahu et al., 2015*) (4 groups of data) assessed acquisition times > 1s. A total of 3670 *in vivo* Raman spectra (cancer = 1,004; normal = 2,666) were acquired from 482 patients. Their coalescent sensitivity and specificity results for fiber-opticRS were 0.87 (95% CI [0.76–0.94], $P = 0.00$, $I^2 = 95.94$) and 0.93 (95% CI [0.90–0.95], $P = 0.00$, $I^2 = 88.50$), respectively. The DOR was 90.13 (95% CI [32.91–246.86], $P = 0.000$, $I^2 = 93.6$). SROC curves were generated to estimate the overall diagnostic accuracy, and the AUC was 0.96 (95% CI [0.94–0.97]).

## Assessment of study quality

All QUADAS-2 items were used to estimate the eligible studies. The risk of bias of the eligible studies is presented in Fig. 2. We can see that all studies were judged as "high risk" on flow and timing domain relating to bias, which is irrational. The reason is that in these 10 studies, all "healthy tissue" has not been performed with pathological examination, while all "cancer tissue" have performed with pathological examination, for the ethical reasons. So, the answer of all the studies is "no" on the signaling question "did all patients receive the same reference standard". Regardless of this issue, most risk assessments were considered "low risk".

## Publication bias and heterogeneity

The forest plot of the sensitivity and specificity of each eligible study is shown in Fig. 3, and indicates that the heterogeneity was significant. In addition, the Q test values of the sensitivity and specificity were 106.23 ($P = 0.00$) and 64.21 ($P = 0.00$), respectively, and the I2 index of the sensitivity and specificity were 85.88 (95% CI [79.99–91.77]) and 76.64 (95% CI [65.45–87.83]), respectively. The results of heterogeneity in each subgroup are presented in Table 3.

No significant publication bias was found in this meta-analysis by Deeks' funnel plot asymmetry test. The funnel plot is shown in Fig. 4.

## DISCUSSION

Currently, there are many technologies that can be used to detect head and neck carcinomas and precancerous lesions. For example, CT, MRI and ultrasound tests are common examinations. And there are other new and approved technologies, for example, confocal microendoscopy, nearinfrared imaging and so on. However, histopathological examination is the only "gold standard" for diagnosis. Although CT/MRI/ultrasound is widely used and is noninvasive, its accuracy in the diagnosis of early precancerous lesions cannot achieve 100% accuracy, and it usually depends on the clinical experience of the doctors, which is subjective. Histological method is invasive and time-consuming, so we hoped to find a noninvasive or minimally invasive, less time-consuming examination to address this issue; in addition, the HE would have high accuracy and specificity. After reviewing the literature, we turned our attention to RS. It has the ability to distinguish different tissues in a noninvasive, real-time manner. Thus, theoretically, RS has the potential to be applied

A

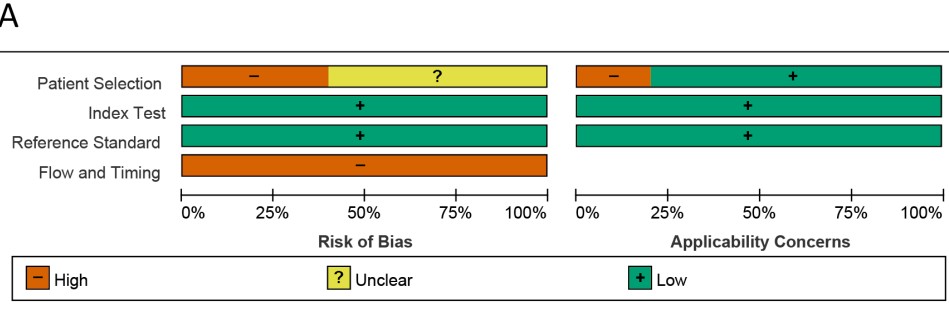

B

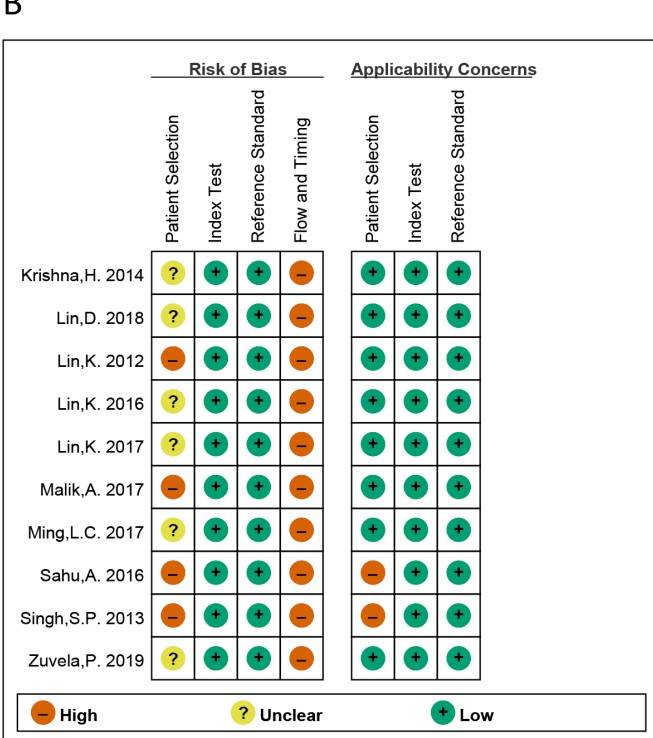

**Figure 2** **The graphical display of the evaluation of the risk of bias and concerns regarding the applicability of the selected studies.** (A) Risk of bias and applicability concerns evaluation of included studies in the pool. (B) Risk of bias and applicability concer.

to clinically distinguish cancer and normal tissue. The fiber optic probe can be applied in the clinic to achieve non-invasive examination. We wanted to know whether fiber-optic RS is reliable in the diagnosis of head and neck carcinomas and to discover its potential in the diagnosis of head and neck carcinomas, so we carried out this analysis.

This meta-analysis assessed the accuracy of fiber-optic RS in the diagnosis of head and neck carcinomas *in vivo* for the first time. A total of ten publications were selected, all of which were published in English. In addition, the relevant research teams were all from Asia, which is explicable because of the high incidence rates of head and neck cancer in Asian countries, such as India and Bangladesh (*Ferlay et al., 2015*; *Hashim et al., 2016*;

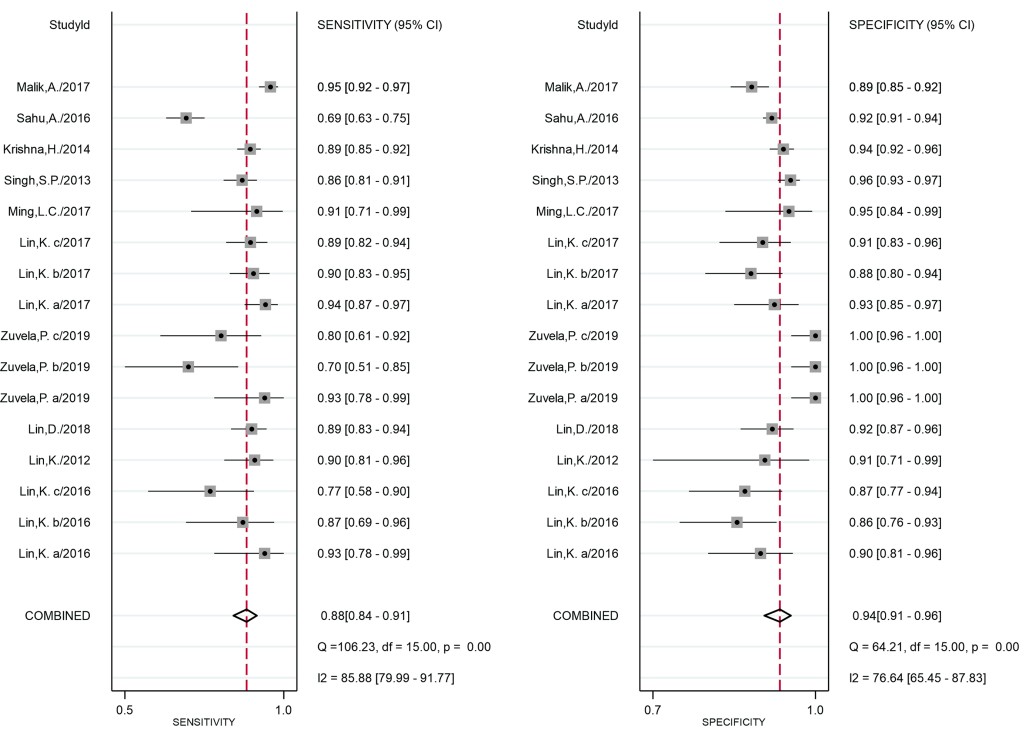

**Figure 3  Forest plot of the sensitivity and specificity of all studies.**

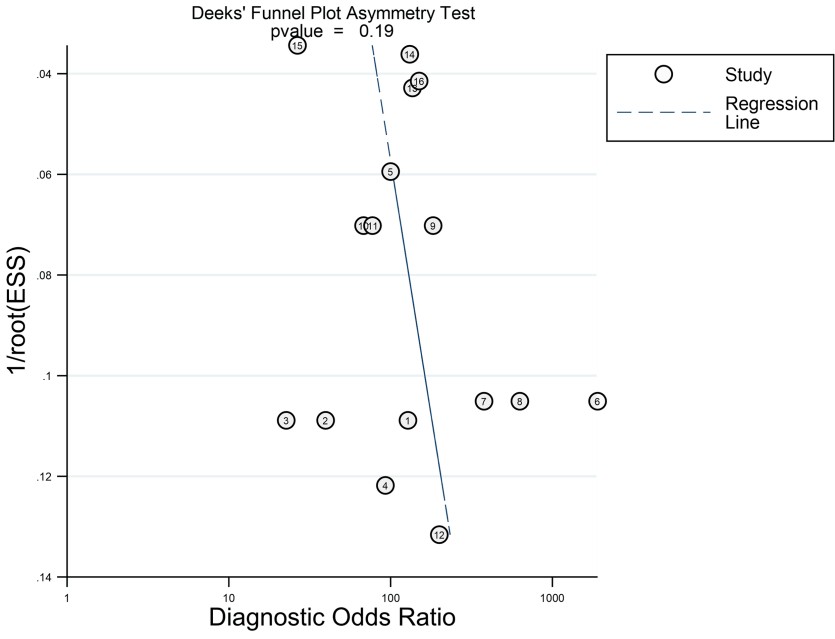

**Figure 4  Deeks' funnel plot asymmetry test.**

**Table 3  Coalescent estimation of sensitivity, specificity, diagnostic odds ratio and area under the curve for fiber-optic Raman spectroscopy.**

| Groups (N) | No. of studies | Groups of data | SEN (95% CI P, I²) | SPE (95% CI P, I²) | DOR (95% CI P, I²) | AUC (95% CI) |
|---|---|---|---|---|---|---|
| All studies | 10 | 16 | 0.88(0.84–0.91) 0.00, 85.88 | 0.94(0.91–0.96) 0.00, 76.64 | 105.69(67.50–165.47) 0.00, 100.00 | 0.96 (0.94–0.97) |
| **Disease position** | | | | | | |
| Larynx Cancer | 2 | 4 | 0.88(0.81–0.92) 0.19, 36.52 | 0.88(0.83–0.92) 0.85, 0.00 | 47.63(21.93–103.45) 0.275, 22.6 | 0.92 (0.89–0.94) |
| Nasopharynx Cancer | 4 | 8 | 0.88(0.83–0.91) 0.02, 57.83 | 0.97(0.90–0.99) 0.00, 72.94 | 118.07(71.38–195.30) 0.294, 17.3 | 0.94 (0.92–0.96) |
| Oral Cancer | 4 | 4 | 0.87(0.76–0.94) 0.00, 95.94 | 0.93(0.90–0.95) 0.00, 88.50 | 90.13(32.91–246.86) 0.000, 93.6 | 0.96 (0.94–0.97) |
| **Raman spectral range** | | | | | | |
| FP | 8 | 8 | 0.87(0.80–0.91) 0.00, 91.70 | 0.93(0.90–0.95) 0.00, 81.45 | 85.36(43.75–166.55) 0.000, 86.1 | 0.96 (0.94–0.97) |
| HW | 4 | 4 | 0.86(0.81–0.91) 0.17, 41.00 | 0.94(0.82–0.98) 0.01, 71.81 | 66.59(24.19–183.29) 0.102, 51.7 | 0.91 (0.88–0.93) |
| FP + HW | 4 | 4 | 0.93(0.88–0.96) 0.97, 0.00 | 0.96(0.88–0.98) 0.04, 63.60 | 199.73(89.96–443.45) 0.493, 0.0 | 0.94 (0.91–0.96) |
| **Acquisition time** | | | | | | |
| ≤1s | 6 | 12 | 0.88(0.85–0.91) 0.03, 49.80 | 0.95(0.90–0.97) 0.00, 67.04 | 92.21(56.49–150.52) 0.101, 36.2 | 0.94 (0.91–0.95) |
| >1s | 4 | 4 | 0.87(0.76–0.94) 0.00, 95.94 | 0.93(0.90–0.95) 0.00, 88.50 | 90.13(32.91–246.86) 0.000, 93.6 | 0.96 (0.94–0.97) |

*Wu et al., 2018*). In addition, for *in vivo* applications, medical device regulations must be followed. These regulations might be stricter outside Asia. Thus, publications from Asian countries were important and necessary for our analysis.

As shown in Table 3, the diagnostic performance of fiber-optic RS for head and neck carcinomas *in vivo* was shown to have with superior specificity and low sensitivity compared to other methods, which was similar to a published meta-analysis (*Zhan et al., 2020*), although the latter measurement was not focused on in vivo. In addition, similar phenomena occurred in the *in vivo* diagnosis of bladder cancer and gastric carcinogenesis (*Chen et al., 2018*; *Bergholt et al., 2013*). Thus, the diagnosis performance of fiber-optic RS *in vivo* for head and neck carcinomas indicate that this method may be more suitable for the confirmation of healthy tissues (*i.e.*, outpatient screening and surgical marginal resection).

To further investigate heterogeneity, subgroup analysis was performed according to sample position, spectroscopy range, acquisition times and sample type. There was no difference between each subgroup in sensitivity, specificity, DOR or AUC, which indicated that fiber-optic RS had stable and reliable diagnostic potential for head and neck carcinomas.

In addition, compared with the use of FP and HW separately, the combination seems to have a tendency to improve sensitivity, specificity and DOR, although there was no

significant difference in the results. It reminds us that more articles are needed to verify this trend.

The FP range contains Raman signals in tissue that indicate specific information, such as proteins, lipids, and deoxyribonucleic acid (DNA) conformations. However, the Raman peak associated with biochemistry in the FP range is quite weak, although the specificity is high (*Lau et al., 2003*; *Huang et al., 2015*), and Raman signals in the FP range may be suppressed because of a weak Raman signal in the tissue and background interference from tissue autofluorescence (AF) (*Lin et al., 2017*; *Lieber & Mahadevan-Jansen, 2003*). In contrast, the HW Raman range includes stronger signals in the tissue with less AF background interference (*Lin, Cheng & Huang, 2012*; *Mo et al., 2009*). *Žuvela et al. (2019)* observed Raman peaks with considerably greater intensity in the HW range. The HW range contains completely different information, such as asymmetric and symmetric $CH_2$ stretching ($\sim$2,885 and $\sim$2,940 $cm^{-1}$) molecules related to proteins and lipids, as well as the water concentration, which may contribute to the development of an *in vivo* Raman spectroscopic diagnostic method (*Lin et al., 2016a*; *Leikin et al., 1997*; *Barroso et al., 2015*). Fiber-optic RS in the combined FP and HW range may have advantages to improve diagnostic performance (*Lin, Cheng & Huang, 2012*; *Mo et al., 2009*; *Bergholt et al., 2016*).

Considering the differences in equipment, subgroups were divided into groups with acquisition times $\leq$ 1 s and acquisition times > 1 s. According to the information in the article, the equipment in the group with acquisition times longer than 1 s generally has the characteristics of the sample's large exposure range, which may lead to inaccurate sample information and ultimately affect the results. Although there is no statistical significance in this result, we believe that uniform equipment conditions are very important and necessary.

Fiber-optic Raman probes are a key component of the translation of RS to *in vivo* clinical applications, and different probe configurations can generate different types of results, leading to inconsistent information concerning the results. In addition, for actual clinical applications in hospitals, the design of fiber optic probes must comply with the basic hospital guidelines: the entire fiber optic spectrum system must be enclosed to avoid stray light and facilitate fiber movement (*Cordero et al., 2018*). Therefore, more advanced research with a large number of samples is required. For the configuration of RS, more information is needed for further research. RS in the FP range and HW range is able to detect differences in malignant tissue at the molecular level with the advantages of being real-time and noninvasive. RS has some limitations in clinical applications. There are some cost and maintenance issues that need to be addressed. For example, fiber-optic RS is very expensive, and the authors are not sure whether hospitals are willing to pay this bill. In addition, the use of fiber-optic RS and the analysis of the results need to be performed in an appropriate place and by professional operators and analysts. Although technical barriers have hindered the translation of RS to *in vivo* clinical applications, fiber-optic RS has exhibited great potential in the diagnosis of head and neck carcinomas with technological improvements (*i.e.*, reduced acquisition time). According to the results of this meta-analysis, fiber-optic RS is an effective method for diagnosing head and neck cancer with high and stable specificity and sensitivity needed to distinguish tumor tissues and nontumor tissues.

We acknowledge that this study still has some limitations. First, the number of included articles and sample size are limited, and most of the sample came from a small number of countries, such as Singapore and India. Therefore, the results and conclusions based on these data are limited, and more clinical studies from more countries are needed to further confirm the utility of fiber-optic RS applications. Second, the heterogeneity of research was very high, which may be due to multiple reasons, such as differences in research teams and inconsistencies in equipment. Third, in the subgroup analysis, the group of oral cancer patients and the group of acquisition times > 1 s included the same data. Thus, we were unable to further analyze sample position. Fourth, the current research has not prospective registration of systematic reviews, but we still strictly followed the steps of systematic evaluation process. Despite all these disadvantages, we are still confident in fiber-optic RS, not only because of its excellent ability to allow users to identify different tissues and components but also because of its excellent accuracy and sensitivity in these limited clinical trials. These clinical trials have shown the tremendous potential of fiber-optic RS in the *in vivo* detection and diagnosis of head and neck carcinomas.

In general, the possibility of fiber-optic RS application in the clinic is high and worthy of further research and development.

## CONCLUSION

*In-vivo* fiber-optic RS is an effective diagnostic tool for head and neck carcinomas. It has high sensitivity and specificity for distinguishing cancerous and healthy tissues. In addition, fiber-optic Raman spectroscopy has great potential and is worthy of further research. Compared with the use of FP and HW separately, the combination seems to have a tendency to improve sensitivity, specificity and DOR, although there was no significant difference. However, considering the high heterogeneity of these studies, more clinical studies are needed to reduce the heterogeneity, and further confirm the utility of fiber-optic Raman spectroscopy *in vivo*.

### Funding

This work was supported by the Natural Science Foundation of Sichuan (2022NSFS1525). The funders had no role in study design, data collection and analysis, decision to publish, or preparation of the manuscript.

### Grant Disclosures

The following grant information was disclosed by the authors:
Natural Science Foundation of Sichuan: 2022NSFS1525.

### Competing Interests

The authors declare there are no competing interests.
## Author Contributions

- Wen Chen conceived and designed the experiments, performed the experiments, analyzed the data, prepared figures and/or tables, and approved the final draft.
- Yafei Chen performed the experiments, analyzed the data, prepared figures and/or tables, and approved the final draft.
- Chenzhou Wu performed the experiments, prepared figures and/or tables, and approved the final draft.
- Xidong Zhang analyzed the data, authored or reviewed drafts of the article, and approved the final draft.
- Xiaofeng Huang conceived and designed the experiments, authored or reviewed drafts of the article, and approved the final draft.

## Data Availability

   The raw measurements are available in the Supplementary Files.

## Supplemental Information

Supplemental information for this article can be found online at http://dx.doi.org/10.7717/peerj.16536#supplemental-information.

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
