# Peer review of "The accuracy of Fiber-Optic Raman Spectroscopy in the detection and diagnosis of head and neck neoplasm in vivo: a systematic review and meta-analysis"

_PeerJ, doi:10.7717/peerj.16536_

## Round 0.1 · original submission · Major Revisions

Please carefully read the comments and suggestions from the reviewers and provide your point-by-point responses.

·

Basic reporting

1- (Major) In the introduction, the authors must mention what is unique about this meta-analysis compared to previously published meta-analyses on this topic (reference no. 7 in this paper and PMID: 35992884). Otherwise, concerns on the originality of this research might arise. After addressing this concern, please revise the statement in line 218-219 and check if it needs an update or not. “This meta-analysis assessed the accuracy of fiber-optic Raman spectroscopy in the diagnosis of head and neck carcinomas in vivo for the first time.”
2- (Major) Authors should add a statement on the data availability in the main text. (are the raw data shared? or Table 2 or in supplementary material or only available with the corresponding author?)
3- Please add a contribution statement as well.
4- The sentence in line 193-195 is incoherent and should be rephrased. “In addition, the Q test values of the sensitivity and specificity were 106.23 (P=0.00) and I2 = 85.88 (95% CI 79.99-91.77), respectively, and the I2 index of the sensitivity and specificity were 64.21 (P=0.00) and I2 =76.64 (95% CI 65.45- 196 87.83), respectively.”

Experimental design

1- (Major) Prospective registration of systematic reviews (for example in PROSPERO) is considered important to prevent authors from repeating undergoing reviews and limit reporting bias (e.g. reporting only significant results). This systematic review did NOT provide details regarding its’ registration. Suggestion: If the authors did not Prospectively register this review, they should explicitly indicate that in the methods section and provide an appropriate justification. Check item #24 of PRISMA checklist to see which details exactly should be mentioned on this part
2- (Major) In line 60, the authors claim that they use the random effect model for analysis. However, this is insufficient, and the authors should provide more details (what module was used STATA: maten? How is the SROC curve created?)
3- Authors should identify the authors who performed the search strategy (within their Methods section).
4- The authors should also describe the screening process (both title/abstract and full-text screening). Please indicate which authors screened, was it blinded or not and mention if a software/website was used in the process.
5- Authors should provide the reasons for excluding articles at the full text screening (preferably in the PRISMA flowchart). This part of item 16b in the PRISMA checklist, which has not been appropriately addressed.
6- If possible, the authors should provide confidence interval for the AUC.

Validity of the findings

1- Authors should expand on the conclusion section and include their own conclusion from the subgroup analysis and discussion.
2- In lines 285-295, the authors discuss some of the limitations of Raman spectroscopy (RS). Those limitations are for RS and not limitations to this study. I suggest that this part becomes a separate paragraph and placed before the meta-analysis limitations section.
3- In the results section (line 186-187), authors mention the process of quality assessment instead of the result of the quality assessment (the included studies was evaluated independently by two reviewers etc.). This part is more appropriate in the methods section. In the results section instead, the authors must mention the results of the quality assessment (overall risk of bias, how many studies were had high risk of bias, which domains of QUADAS-2 were mostly affected, etc.)

Additional comments

1- Abbreviations use has been inconsistent. For example, line 18 Raman spectroscopy abbreviation “RS” was not introduced and has been use inconsistently thereafter.
2- Table 2 lacks the appropriate cation that includes the abbreviations used within it.
3- Figure 1 shows records after duplication removed are 324. However, it shows that only 86 has been screened. Authors should fix the PRISMA flowchart and make it more coherent.
4- Figure 2A is missing the labels for the 3 bars on the right side (patient selection, index test, reference standard).
5- Figures are of poor quality. Hopefully the authors can provide clearer figures.

·

Basic reporting

no comment

Experimental design

no comment

Validity of the findings

no comment

Additional comments

no comment

Reviewer 3 ·

Basic reporting

The article describes a literature based review of the applications of fiber-based Raman spectroscopy for the detection and diagnosis of head and neck neoplasm, in vivo. The rationale, methodology, and outcomes of the study are well argued, presented and justified.

Experimental design

The Review design and methodology is well described and justified, including inclusion and exclusion criteria, and refinement of the scope from the initial search.

Validity of the findings

The findings are clearly and validly presented, without bias.

Additional comments

(i) In the Abstract, and at the end of the Introduction, the authors should reconsider the statement:
"The aim of this article was to systematically evaluate the diagnostic accuracy of ûber-optic Raman spectroscopy (RS)...". The aim is rather to review and collectively assess the published studies of the in vivo detection and diagnosis of head and neck carcinomas, and to derive a consensus average of the accuracy, sensitivity and specificity.
(ii) At the end of the abstract, the authors should further qualify their statement "However, more clinical studies are needed to further confirm the utility of fiber-optic Raman spectroscopy in vivo.", (for example "to reduce the heterogeneity of these studies".
This qualifying recommendation from the abstract should also vbe expressed in the Discussion/Conclusions of the overall acrticle.
.

---

## Round 0.2 · Minor Revisions

Some minor comments have been raised by Reviewer 1 and should be addressed.

In comment number 2, what I meant by data availability is to add in the declaration section at the end of the paper “data availability”. For example, “The raw measurements are available in the supplementary files 1.”

Please upload your collated dataset used for the metaanalysis to the PeerJ system.

It is unfortunate that you cannot fix the labels in Figure 2.

In the abstract, I believe that the “196” in “76.64 (95% CI 65.45- 196 87.83)” is a typo and should be removed. The same typo is present in 3.4 Publication bias and heterogeneity section.

·

Basic reporting

Dear authors, thank you for taking the time and effort to address my comments.

In comment number 2, what I meant by data availability is to add in the declaration section at the end of the paper “data availability”. For example, “The data used for analysis in this work is available upon reasonable.”

It is unfortunate that you cannot fix the labels in Figure 2.

In the abstract, I believe that the “196” in “76.64 (95% CI 65.45- 196 87.83)” is a typo and should be removed. The same typo is present in 3.4 Publication bias and heterogeneity section.

Experimental design

The revised article now meets the required standards in experimental design.

Validity of the findings

The revised article now meets the required standards in the validity of its findings.

Additional comments

No additional comments.

·

Basic reporting

no comment

Experimental design

no comment

Validity of the findings

no comment

Additional comments

no comment

---

## Round 0.3 · accepted · Accept

The authors have addressed the reviewers' comments.